# Evaluating Change of Marginal Bone Height with Cone-Beam Computed Tomography Following Surgical Treatment with Guided Tissue Regeneration (Bone Grafting) or Access Flap Alone: A Retrospective Study

**DOI:** 10.3390/medicina57090869

**Published:** 2021-08-25

**Authors:** In-Kyung Lee, Hyun-Seok Choi, Sang-Heon Jeong, Jung-Tae Lee

**Affiliations:** 1Department of Periodontics, Jukjeon Dental Hospital, College of Dentistry, Dankook University, Yongin 16890, Korea; perio8296@dankook.ac.kr (I.-K.L.); futurechoi0621@gmail.com (H.-S.C.); 2Department of Radiology, Jukjeon Dental Hospital, College of Dentistry, Dankook University, Yongin 16890, Korea; serch34@hanmail.net; 3Department of Periodontics, One-Stop Specialty Center, Seoul National University, Dental Hospital, Seoul 05698, Korea

**Keywords:** periodontitis, dental materials, guided tissue regeneration, osseous defects, periodontal regeneration, periodontal surgery

## Abstract

*Background and Objectives:* This study aimed to evaluate the change of bone height following treatment of human intrabony defects with guided tissue regeneration (GTR) with bone grafting or access flap alone by cone-beam computed tomography (CBCT) scan. *Materials and methods:* This study was conducted as a retrospective longitudinal study. In this study, a total of 2281 teeth sites were included: the GTR group had 1210 sites, and the Flap group had 1071 sites. In the GTR group, demineralized freeze-dried bone (DFDBA) particles in combination with resorbable collagen membrane were used. No regenerative material was applied to the Flap group. CBCT images were taken twice at baseline and at least 2.5 months postoperatively. Bone heights were measured using software on CBCT images. *Results:* The bony change between the GTR and Flap groups was significantly different (*p* = 0.00001). Both males and females in the GTR group had smaller bone loss than in the Flap group. In age groups, significant differences of bony height between the GTR and Flap groups were observed in the subgroups consisting of those 29–45 and 46–53 years old. The non-smoking subjects in the GTR group had higher bone heights than those in the Flap group. In the absence of systemic disease and medicine, bone formation was higher in the GTR group than in the Flap group. In terms of oral position, the #14–17, #34–37, and #44–47 subgroups of the GTR group showed higher levels of bone heights than those of the Flap group. *Conclusions.* The results of this study indicated that the GTR procedure offers the additional benefit of higher bone heights than the Flap procedure does.

## 1. Introduction

The main objective of periodontal surgery is to contribute to the long-term preservation of the periodontium by facilitating plaque removal and infection control. Periodontal surgery is divided into open flap surgery and guided tissue regeneration (GTR). Open flap surgery aims to reconstruct the periodontal pocket by removing inflamed tissue and subgingival calculus. Open flap surgery provides the following: (1) accessibility for proper professional scaling and root planing; (2) establishing a gingival morphology that facilitates self-performed infection control; and (3) creating new attachment in the destructive site of the gingiva [1]. GTR is the regenerative procedure using bone graft material and barrier membranes [2,3]. GTR has the advantage of mechanical reinforcement of bony defects with bone graft material and membranes, thus providing space maintenance during healing periods. In the previous study, GTR has been shown to result in significantly more periodontal regeneration [4]. Sculean et al. and Stavropoulos et al. investigated that GTR had more increased clinical attachment level (CAL) gains compared with flap alone in clinical studies [5,6]. These approaches (GTR or open flap surgery), while offering tangible benefits, remain inconclusive. Few cases did enough to provide evidence of which procedure is better.

Demineralized freeze-dried bone (DFDBA) particles have been used in combination with barrier membrane in GTR, resulting in enhanced periodontal regeneration due to the osteoinductive ability of DFDBA to stimulate bone formation. DFDBA is associated with the amount of bone morphogenic proteins that remain after the demineralization process is completed [7,8]. A previous study demonstrated that probing pocket depth (PPD) and CAL gain were observed in sites treated with a combination of DFDBA and barrier membrane [9]. Similarly, Kiany and Moloudi reported favorable results after using DFDBA and barrier membrane [10].

A barrier membrane should have five characteristics to support ideal function: biocompatibility, space-maintenance ability, tissue integration, cell occlusiveness, and ease of manipulation [11]. Resorbable membrane has been used in GTR due to its biocompatibility [12]. However, resorbable membrane has weak physical properties. To maintain infrabony defect with bone graft material, resorbable membrane needs to be strengthened. A cross-linking procedure has been introduced that increases mechanical property and slows the degradation of resorbable membranes [13]. A cross-linking technique of resorbable membranes has two classified methods: physical and chemical methods. Physically cross-linked membranes were manufactured using ultraviolet (UV) irradiation or a dehydrothermal (DHT) technique. Chemically cross-linked membranes were made by glutaraldehyde (GTA) and hexamethylene diisocyanate (HMDI) [14]. A previous study demonstrated that cross-linked membranes overcome a shortage of non-cross-linked membranes [15].

To gain accurate results of periodontal treatment, various measurement methods have been developed. Clinical methods such as bone sounding are not always able to provide accurate tissue measurement. Surgical reentry and histologic evaluation are prohibited due to ethical issues. Therefore, radiographic methods (intraoral periapical and panoramic radiographs) were used to evaluate differences in marginal bone loss before and after surgery. The limitation of conventional radiographic measurement is that it cannot provide multi-dimensional assessment. Cone-beam computed tomography (CBCT), a reliable and widely used tool, provides three-dimensional exploration of periodontal osseous defects and assesses the furcation area of molars [16,17,18].

The aim of this study was to evaluate the change of bone height following treatment of human intrabony defects with GTR (bone grafting) or access flap alone.

## 2. Materials and Methods

### 2.1. Study Population

The study was conducted as a retrospective longitudinal study. The study was conducted according to the guidelines of the Declaration of Helsinki and approved by the Institutional Review Board for Clinical Research at Dankook University College of Dentistry Jukjeon Dental Hospital (approval no. 201910-001-002). From May 2015 to May 2018, 178 patients with 2520 sites (GTR: 97 patients (1296 sites); Flap: 81 patients (1224 sites)) who visited the department of periodontics at Dankook University College of dentistry Jukjeon Dental Hospital for periodontal treatment were selected. All patients met the inclusion criteria as follows by referring to previous studies [19,20,21].

Clinical and radiographical diagnosis of severe chronic periodontitis;Presence of at least one intrabony defect of ≥ 3 mm on the radiographs;PPD ≥5 mm in the intrabony defects.

Written informed consent was obtained from all the patients. Each site designated mesial and distal aspects on a panoramic view of CBCT images. After sorting due to scattering of CBCT image and extraction (28 teeth (56 sites) in Flap group) during the follow-up period, we divided the 2 groups by site (GTR: 1210 sites and Flap: 1071 sites). Premolars and molars of maxilla and mandible were included. Study participants were excluded if they did not complete periodontal treatment, had teeth extracted during the follow-up period, had no CBCT images both before and after treatment, and had <5 mm PPD, which is not suitable for periodontal surgery (GTR or Flap) (Figure 1). Sample size calculation in this study referenced previous similar research by Tonetti et al. [20]. The power of 0.8 and an alpha level of 0.05 were used to detect the difference between groups.

### 2.2. Pre-Treatment

Scaling or root planing was performed on all participants prior to the experimental phase.

### 2.3. Surgical Procedures

One periodontist (JTL) performed all surgical procedures. The patients received oral hygiene instructions before periodontal surgery. Under local anesthesia using 2% lidocaine (1:100,000 epinephrine, Huons Co., Ltd., Sungnam, Korea), the surgical sites were treated. In the GTR group, a full thickness flap was performed. The exposed defects were carefully scaled and root planed using a combination of mechanical and hand instruments. DFDBA (Human Cortical Powder, Demineralised, DIZG, Berlin, Germany) was applied to the defect area. A cross-linked-collagen membrane (OssGuide, Hyundai bioland Co., Ltd., Cheongju, Korea) composed of porcine pericardium-derived type I collagen was used. It was trimmed to the local anatomy and then positioned on the graft material. Then, the flap was replaced and sutured with resorbable sutures (4-0 Vicryl, Ethicon, Somerville, NJ, USA). All patients were prescribed amoxycillin and clavulanic acid 375 mg (Augmentin, Ilsung Pharma Co., Seoul, Korea), naproxen sodium (Anaprox, Jongeun Dang Pharmceutical, Co., Seoul, Korea), and almagate (Almagel, Yuhan Pharma Co., Seoul, Korea) 3 times daily for 1 week unless an allergy to penicillin was present. All patients were instructed to rinse for 30 seconds twice daily with 0.12% chlorhexidine gluconate (Hexamedin, Bukwang Pharmaceutical, Ansan, Korea) for 1 week. Sutures were removed 2 weeks post-surgery. In the Flap group, full thickness flap, scaling, and root planing were performed. Irregular bony protrusion was removed with rotary or hand instruments. After debridement, the flap was repositioned and sutured. The same medicines were prescribed as in the GTR group (Figure 2).

### 2.4. CBCT Taking and Maintenance Care (3 and 6 Months)

CBCT images were taken twice at baseline (before periodontal surgery) and at least 2.5 months postoperatively. A supportive care program and professional calculus removal were provided to all patients at 3 and 6 months. After that, regular oral cleaning was performed every 6 months if the patients maintained their oral hygiene.

### 2.5. Radiographic Evaluation and Bone Height Measurement

A CBCT scanner (Kodak 9500, Carestream Health, Rochester, NY, USA) was used in this study, providing a grayscale image of 14 bits with a voxel size of 0.2 mm per side. The CBCT images were viewed using 3D imaging software (OnDemand 3D, Cybermed Co., Seoul, Korea). In order to evaluate change of bone height, CBCT images were measured twice before and after treatment on the same site. The cemento-enamel junction (CEJ) was set as an unchanged reference [16]. If an implant was involved, the reference point was the connection between the abutment and the crown. In the preoperative CBCT image, the distance of the CEJ-base of alveolar bone of the mesial and distal sites of the teeth was measured (Figure 3A–D). To overlap the images of the same cross-section as much as possible before and after treatment, imaginary lines were used (Figure 3E; connecting line of CEJ, A, and D: bone height of first CBCT image, Figure 3F: connecting line of basic point between Figure 3A,D). The distance of the same site after treatment was measured on the second CBCT image (Figure 3A’–D’). The amount of bony change was confirmed by subtracting the first values from the second values (Figure 3A’–D’,A–D). To attest the consistency of the measurements, each section of the CBCT image was assigned a serial number. All pre- and postsurgical bone height measurements were carried out by a single examiner (H.S.C.). A second examiner (J.T.L.) also evaluated bony height at the same serial number image.

### 2.6. Statistical Analysis

Statistical analysis was performed using SPSS software (SPSS version 23.0, Chicago, IL, USA). To compare variables between GTR and Flap groups, a Mann–Whitney U test was used. Kruskal–Wallis was used to compare variables in 3 or more groups including smoking, systemic disease, medicine, oral regions, and precise site. The threshold for statistical significance was 5%. The post hoc Bonferroni correction was used for multiple comparisons between groups. A logistic regression model was used for multivariable analysis.

## 3. Results

### 3.1. Baseline Characteristics

Baseline demographics are summarized in Table 1. There was a significant different bony change between the GTR and Flap groups (*p* = 0.00001). A total of 1210 subjects of GTR (594 (49.1%) male and 616 (50.9%) female) and 1071 subjects of Flap (666 (62.2%) male and 405 (37.8%) female) were included. Both male and female in the GTR group had smaller bone loss than in the Flap group (*p* = 0.010 and *p* = 0.001, respectively). The mean age was 49.8 ± 8.0 (GTR) and 51.3 ± 9.4 (Flap). Significant differences in bony height between GTR and Flap were observed in the 29–45 and 46–53 subgroups (*p* = 0.00001 and *p* = 0.040, for each). In GTR group, the 29–45 subgroup had significantly favorable results compared with the 46–53 and 54–76 subgroups (*p* = 0.0004 and *p* = 0.006, for each). The non-smoking subjects in the GTR group had higher bone heights than those in the Flap group (−0.09 ± 1.13; GTR and 0.12 ± 0.93; Flap, *p* = 0.009). The average intervals between the first and second CBCT was 692.7 ± 351.3 days in the GTR group and 688.4 ± 342.9 days in the Flap group. All values of CBCT interval were higher in the GTR subgroups than in the subgroups in Flap (87–490 days: −0.06 ± 1.23, 491–859 days: 0.02 ± 1.24: 860–1543 days: 0.11 ± 1.40 vs. 0.17 ± 0.83, 0.20 ± 1.24, 0.20 ± 0.95). Two subgroups (87–490 and 491–859) had significant differences of bone healing in GTR and Flap (*p* = 0.0001 and *p* = 0.018, respectively). With systemic diseases, there was an association between mean heights of alveolar bone of GTR and Flap in hepatitis. Bone height for the Flap group was higher than in the GTR group with significant difference (0.32 ± 0.84: GTR vs. −0.35 ± 0.74: Flap, *p* = 0.022). No difference was found in other diseases, including hypertension, rhinitis, sinusitis, diabetes mellitus, and hyperlipidemia. In the absence of systemic disease, bone formation was higher in the GTR group than in the Flap group (0.02 ± 1.31: GTR vs. 0.22 ± 1.05: Flap, *p* = 0.00001). In subjects taking medications for diabetes mellitus and hyperlipidemia, alveolar bone recovery was less in the GTR group than in the Flap group (0.23 ± 0.51: GTR vs. −0.21 ± 0.96: Flap, *p* = 0.018). However, there was a significantly higher difference for height of bone in the GTR group without medicine than in the Flap group (0.03 ± 1.31: GTR vs. 0.21 ± 1.05: Flap, *p* = 0.00001). According to oral position, the #14–17, #34–37, and #44–47 subgroups of the GTR group showed higher levels of bone heights than those of the Flap group (0.04 ± 1.21, −0.03 ± 1.10, and −0.03 ± 1.29; GTR vs. 0.32 ± 0.91, 0.18 ± 0.91, and 0.18 ± 1.25; Flap, *p* = 0.00001, 0.024, and 0.087, respectively). Only the #24–27 group had reverse results of other oral sites (0.10 ± 1.50; GTR vs. 0.07 ± 1.05; Flap, *p* = 0.342). In all premolars and molars in the GTR group, bone height was higher than that of the Flap group. In particular, it was significantly higher at #4, #5, and #7 (*p* = 0.033, 0.002, and 0.039, for each) (Figure 4).

### 3.2. Association of Tooth Site and Bone Height between GTR and Flap

The bony height values of #14D, #15M, and #47M in the GTR group were significantly higher than those in the Flap group (*p* = 0.016, *p* = 0.036, and *p* = 0.023, for each). In addition, the bony height values of #14M, #17D, and #35M in the GTR group were close to significance (*p* = 0.068, *p* = 0.060, and *p* = 0.073, respectively). However, the change of bony height value for #25D in the Flap group revealed a significant difference to that in the GTR group (*p* = 0.043) (Table 2).

### 3.3. Association of Sex and Bone Height between GTR and Flap

In the GTR group, women were more effective at #14D and #15M (*p* = 0.028 and *p* = 0.009, for each). In males, GTR increased bone resorption at #15D. In males, the GTR group had increased bone resorption at #15D compared with the Flap group (0.54 ± 1.32; GTR vs. 0.25 ± 0.85; Flap, *p* = 0.635). In the case of #24D, the amount of bony change of GTR group was smaller than that of the Flap group regardless of gender (Male: 0.11 ± 0.77; GTR vs. −0.15 ± 0.92; Flap, *p* = 0.315, Female: −0.23 ± 0.85; GTR vs. −0.09 ± 0.79; Flap, *p* = 0.594). Both males and females showed better bony healing at #35 M, but the GTR group was especially effective in females (Male: −0.18 ± 1.22; GTR vs. 0.06 ± 0.68; Flap, *p* = 0.542, Female: −0.27 ± 0.45; GTR vs. 0.40 ± 0.66; Flap, *p* = 0.012). Comparison of GTR and Flap in males showed significant results at #47M (−0.77 ± 1.32; GTR vs. 0.39 ± 1.89; Flap, *p* = 0.017) (Table 3).

### 3.4. Association of Smoking and Bone Height between GTR and Flap

The GTR and Flap groups differed by smoking status. There was a significant difference between the two groups in non-smoking (*p* = 0.009). Among smokers, 10 of 32 sites in the GTR group showed less bone healing than in the Flap group (#14M, #16D, #24M, #24D, #25M, #27M, #27D, #34M, #44D, and #46D). In non-smokers, the bone healing of the #24M and #34 sites was better in the GTR group (*p* = 0.011 and *p* = 0.019, respectively) (Table 4).

### 3.5. Association of Age and Bone Height between GTR and Flap

At 29–45 years of age, 24 sites showed better bone recovery in the GTR group than in the Flap group. Four sites (#14D, #35M, #47M, and #47D) in the GTR group showed significant differences (*p* = 0.016, *p* = 0.036, *p* = 0.010, and *p* = 0.011, for each). The 46–53 subgroup showed less bone resorption and bone filling in the GTR group than in the Flap group. In the 54–76 subgroup, the number of favorable healing sites in the GTR group was less than in the sites of the other two subgroups. No difference was found in the 54–76 subgroup (Table 5).

### 3.6. Association of CBCT Interval and Bone Height between GTR and Flap

The smaller the interval between before and after CBCT, the more bone increase and the less bone resorption. In particular, in the 860–1543 days group, there was a significant difference in bone growth or small bone resorption at #14D, #26M, and #35M (*p* = 0.048, *p* = 0.032, and *p* = 0.042, respectively) (Table 6).

### 3.7. Multivariable Analysis for Alveolar Bone Loss after Treatment

In multivariable analysis for alveolar bone loss after treatment, the type of surgery and upper/lower jaw variables were significantly associated with alveolar bone loss (odds ratio (OR), 0.731; 95% confidence interval (CI), 0.619–0.862; *p* = 0.001; type of surgery vs. odds ratio (OR), 1.255; 95% confidence interval (CI), 1.063–1.481; upper/lower jaw, *p* = 0.007) (Table 7).

## 4. Discussion

The present study showed that the change of bone height after GTR and after access flap alone procedures showed differences. Our findings indicate that the bone height of several teeth in the GTR group was significantly higher compared with those teeth in the Flap group (#14D, #15M, and #47M). Females in the GTR group had significantly more favorable values than males. In males, #47 showed significant healing of bone height. Regarding smoking, there was notable bone loss in the GTR group among smokers. All patients were divided by three age-related subgroups consisting of those 29–45, 46–53, and 54–76 years old. At lower ages (29–45), the GTR group showed less bone resorption than the Flap group. The smaller the interval between before and after CBCT, the more bone increase and the less bone resorption. It was observed that the smaller the interval between shots of CBCT, the less bone resorption. In particular, in the absence of systemic diseases or no medications, the GTR group showed less bone resorption and better healing than the Flap group. A multivariable analysis suggested that the type of surgery and surgical site (upper/lower jaw) were associated with bone resorption after treatment.

In this study, the comparison between groups showed less alveolar bone loss and superior bone healing in the GTR group vs. the Flap group as a whole. The same result was seen in a previous study. Tonetti et al. suggested that regenerative periodontal surgery with GTR had PPD reduction and CAL gain [20]. They also noted that the absolute value of the added healing portion was relatively small, but it was in agreement with other similar research studies [22,23]. In a systemic review of 13 articles, regenerative surgeries have shown an adjunctive benefit of CAL gain [24]. A distinctive point observed in this study was that 28 teeth were extracted only from the Flap group during the follow-up period after periodontal surgery. It seems that the GTR procedure is a superior treatment to prolong tooth life span compared with the Flap procedure. Long-term study of GTR vs. open flap reported that there was no significant difference of bone loss between GTR and open Flap surgery [21]. The present study revealed that GTR in the mandible yielded better bone gain than in the maxilla. These outcomes are similar to those reported by Odontuya et al. [25]. Other researchers explained that this difference was due to the complex morphology and accessibility of molars such as root cuvature and furcations [26,27].

This study used DFDBA in combination with resorbable membrane in the GTR group. DFDBA is an allograft with potential osteoinductive ability to expose BMPs that promote osteoblast differentiation, and it would be beneficial for bone regeneration [28,29]. Camelo et al. demonstrated that DFDBA with collagen membrane had better clinical outcomes of bony healing in a human study [30]. In an animal study, the combination of membrane and DFDBA has been proven to regenerate bone [31]. By demineralized processing, DFDBA has higher osteoinductivity but lower mechanical property. Bone graft material should prevent the resorbable membrane from moving downward. For this reason, physical strength is necessary. DFDBA has relatively weaker mechanical properties than freeze-dried bone allograft (FDBA) or xenograft, and it might affect results when used clinically. A previous study failed to find a difference between bone grafting with DFDBA and a no-bone grafting procedure [32].

Resorbable membranes have been introduced as a barrier to prevent moving epithelium apically [33]. While resorbable membranes have many favorable aspects, they have major limitations, including weaker physical strength [34]. A previous study found that it takes approximately 4–24 weeks for the degradation of collagen membranes [11]. Cross-linked membranes have been developed to increase the stiffness and delay the degradation of resorbable membranes and have shown superior physical strength relative to non-cross-linked membranes [13,35]. Cross-linked membranes were used in this study. In a previous study, the application of resorbable membranes was favorable at infrabony defects [32]. Another study demonstrated that combining bone graft material with resorbable membrane resulted in more benefit than only membrane [36]. However, Stavropoulos et al. did not find any difference between those two methods (bone + membrane vs. membrane only) [6]. In this study, porcine collagen was used as the source of resorbable membranes. Bovine collagen is one of the major sources of resorbable membranes, but the drawback of this collagen is the associated allergic reactions [37]. Porcine collagen in a known alternative to bovine collagen. A previous study found that porcine collagen led to minimal allergic reactions because porcine collagen seemed more similar to human collagen [38]. Lee et al. reported a similar clinical expediency and properties when comparing porcine cross-linked vs. non-cross-linked collagen membrane [39].

This study differs from previous studies in that it compared the situation before and after treatment using CBCT. The detection of periodontal bone loss is mandatory for accurate diagnosis. Clinical methods including probing have shown limitations in reliability. Various factors such as the probing force, shape of the tip, and direction of the probe affect the clinical results. In addition, it is not easy to accurately assess the healing point due to resistance of tissue [40]. It is difficult to get an accurate image using overlapping images from conventional radiographs such as intraoral periapical and panoramic radiographs. CBCT has been introduced to overcome the drawbacks of conventional radiographic methods. CBCT can get a specific cross-sectional image and can reconstruct radiographic images with a multi-dimensional view [16,17,18]. There were several previous studies in which CBCT was used to measure the amount of bone loss in periodontitis. Mohan et al. reported that there was no difference in bone loss between CBCT and actual measurement of surgically exposed osseous defects [40]. In addition, CBCT was useful for identifying buccal and lingual bony defects in aggressive periodontitis. Another study has concluded that CBCT had the ability to assess the maxillary or mandibular furcation area [41]. CBCT can identify the root concavities of premolars and bone loss pattern [42]. An important advantage of CBCT is the low radiation dose. The radiation of CBCT has been reported to be 15 times lower than conventional radiography [43]. 

The limitation of this restorative study is that the conditions such as systemic disease and oral environment of the test and control group could not be perfectly matched. To enhance these drawback, future studies will investigate prospective research for the controlled subjects. In addition, other bone substitutes including xenograft and alloplast might be suggested to use for further research. A strength of the present study is that a large number of subjects using CBCT were involved. The design of this study differs from previous studies in that it directly observed bone changes using CBCT. PPD and CAL may be good clinical indicators. However, radiologic measurement with CBCT is also good and accurate for evaluating the effectiveness of periodontal treatment.

## 5. Conclusions

Despite the limitations of this study, the results suggest that GTR with bone grafting can be a more effective method for bone healing and delay the extraction of teeth than Flap alone.

## Figures and Tables

**Figure 1 medicina-57-00869-f001:**
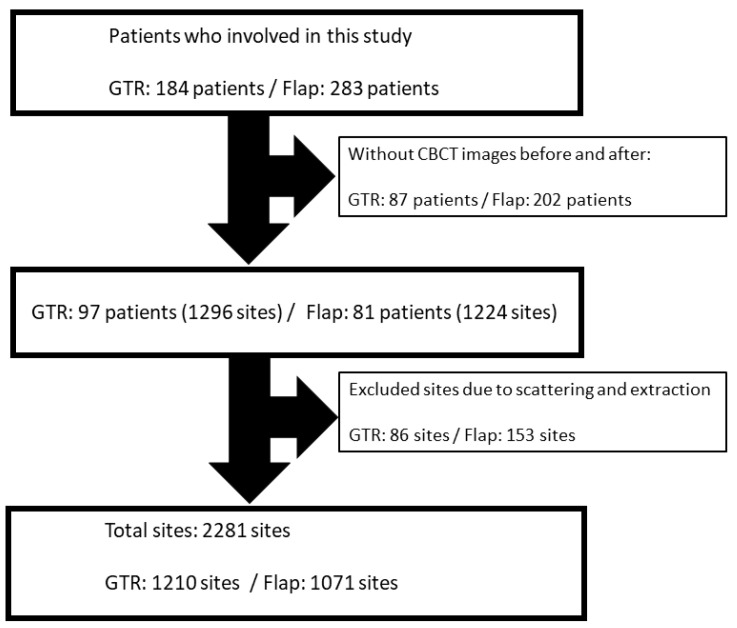
A flow diagram of study participants’ enrollment and follow-up. GTR: guided tissue regeneration; CBCT: cone-beam computed tomography.

**Figure 2 medicina-57-00869-f002:**
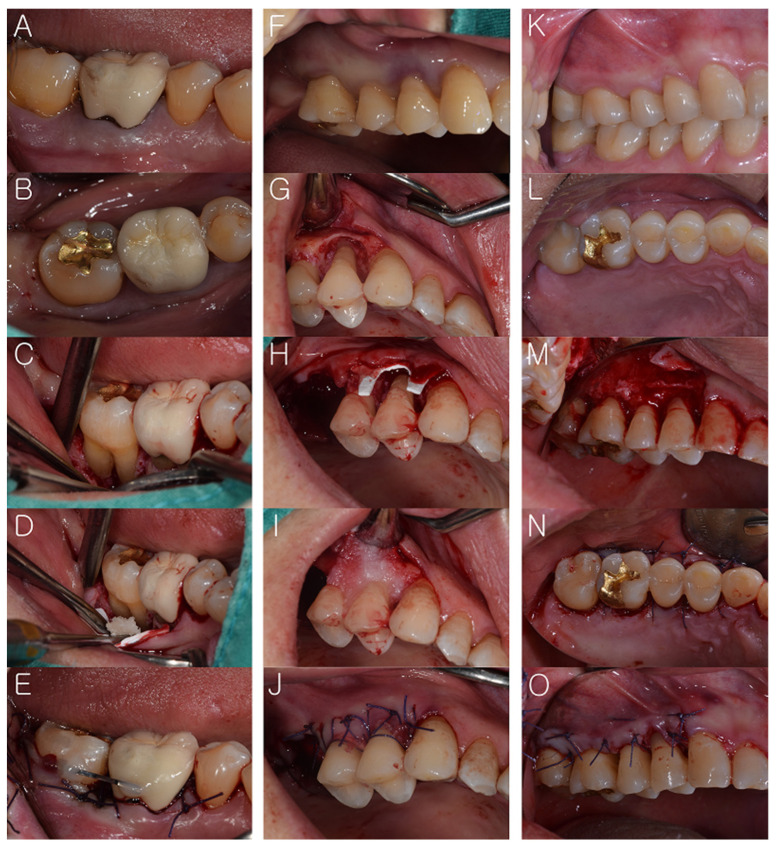
(Case 1: **A**–**E** and Case 2: **F**–**J**): Guided tissue regeneration (GTR); (**A**–**B**,**F**) before treatment, (**C**,**G**) full thickness flaps were performed; then, the exposed defects were carefully scaled and root planed, (**D**,**H**–**I**) demineralized freeze-dried bone (DFDBA) and collagen membrane were applied to the defect area, (**E**,**J**) suture and resin wire splint were performed. (Case 1: **K**–**O**): Open flap surgery (Flap); (**K**,**L**) before treatment, (**M**) full thickness flap and debridement were also performed, (**N**,**O**) suture was performed.

**Figure 3 medicina-57-00869-f003:**
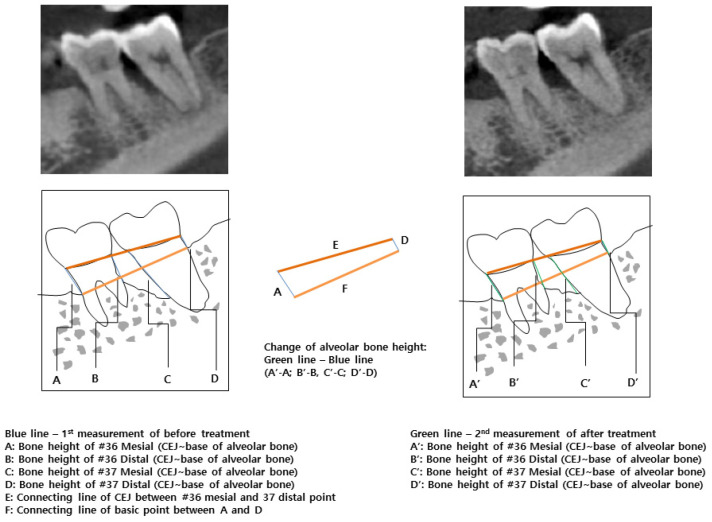
(**A**–**D**): The distance between the cemento-enamel junction (CEJ) and base of alveolar bone of the mesial and distal sites on the teeth in the preoperative CBCT image, (**E**): Connecting line of CEJ, (**F**): Connecting line of basic point between (**A**) and (**D****)**, (**A**’–**D**’): The distance of the same site on the second CBCT image after treatment, (**A**’–**D**’): The amount of bony change.

**Figure 4 medicina-57-00869-f004:**
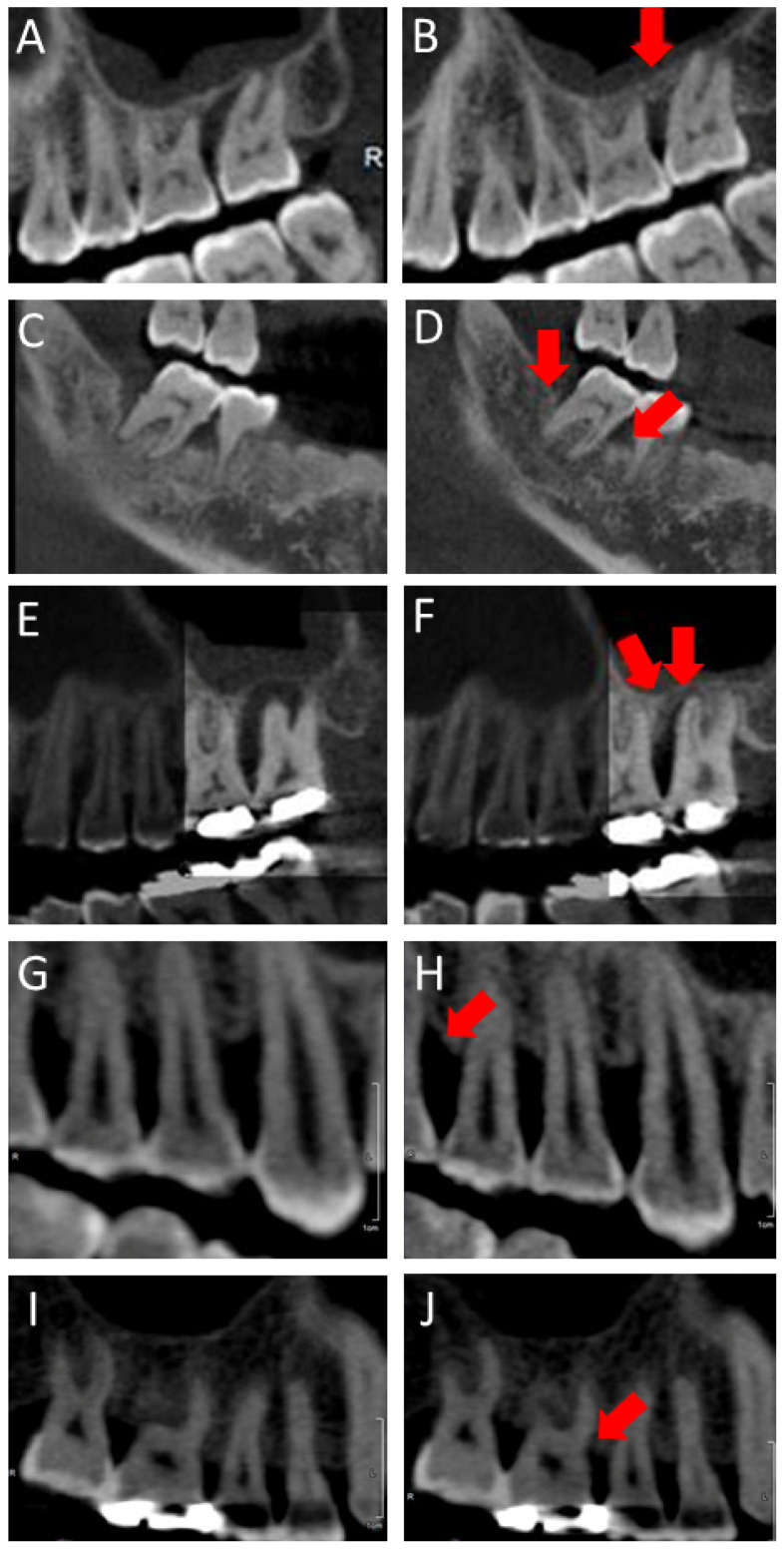
(**A**–**F**): CBCT images before and after treatment in the GTR group. (**G**–**J**): CBCT images before and after treatment in the Flap group (Arrow: bone healing area).

**Table 1 medicina-57-00869-t001:** Baseline characteristics.

Variable	GTR (mm)			Flap (mm)			*p* 2
	*N*	Mean (SD)	*p* 1	*N*	Mean (SD)	*p* 1	
Total N/mean (SD)	2281			0.10 (1.19)			
	1210	0.03 (0.13)		1071	0.190 (0.104)		0.00001 *
Sex			0.939			0.079	
Male (%)	594 (49.1)	0.01 (1.22)		666 (62.2)	0.18 (1.14)		0.010 *
Female (%)	616 (50.9)	0.05 (1.37)		405 (37.8)	0.20 (0.85)		0.00002 *
Age [years, mean (SD)]		49.77 (8.03)	0.001 *		51.25 (9.40)	0.417	
29–45 (%)	416 (34)	−0.17 (1.21)		376 (35)	0.17 (0.10)		0.00001 *
46–53 (%)	388 (32)	0.19 (1.26)	0.0004 *	312 (29)	0.24 (1.11)	0.217	0.040 *
54–76 (%)	406 (34)	0.08 (1.39)	0.006 * 0.565	383 (36)	0.16 (1.01)	0.901 0.280	0.125
Smoking			0.551			0.301	
Present (%)	280 (21.6)	0.10 (1.30)		239 (19.5)	0.22 (1.23)		0.120
Past (%)	91 (7.0)	−0.05 (1.05)	0.415	82 (6.7)	0.09 (1.14)	0.218	0.693
Not (%)	839 (64.7)	0.01 (1.32)	0.317 0.930	750 (61.3)	0.12 (0.96)	0.891 0.334	0.009 *
CBCT interval [days, mean (SD)]		692.67 (351.28)	0.131		688.42 (342.93)	0.954	
87–490	404 (33)	−0.06 (1.23)		332 (31)	0.17 (0.83)		0.0001 *
491–859	362 (30)	0.02 (1.24)	0.241	419 (39)	0.20 (1.24)	0.795	0.018 *
860–1543	444 (37)	0.11 (1.40)	0.051 0.373	320 (30)	0.20 (0.95)	0.948 0.797	0.092
Systemic diseases			0.546			0.004 *	
Hypertension (%)	74 (6)	0.15 (1.49)		140 (13)	0.30 (1.17)		0.130
Rhinitis, Sinusitis (%)	14 (1)	−0.01 (0.47)	0.986	64 (6)	0.03 (0.82)	0.104 0.083	0.824
Hepatitis (%)	14 (1)	0.32 (0.84)	0.201 0.129	16 (1)	−0.35 (0.74)	0.021 0.013 0.145	0.022 *
Diabetes mellitus, Hyperlipidemia (%)	30 (2)	−0.15 (0.72)	0.504 0.648 0.051	68 (6)	−0.05 (0.84)	0.017 0.007 0.411 0.335	0.945
None (%)	1078 (89)	0.02 (1.31)	0.963 0.905 0.132 0.399	783 (73)	0.22 (1.05)	0.619	0.00001 *
Medicine			0.286			0.011 *	
Aspirin, Hypertension (%)	30 (2)	−0.22 (0.98)		148 (14)	0.17 (1.02)		0.107
Anticoagulant (%)	32 (3)	−0.01 (1.44)	0.972	0 (0)	-	-	-
Diabetes mellitus, hyperlipidemia (%)	24 (2)	0.23 (0.51)	0.084 0.071	40 (4)	−0.21 (0.96)	0.023 0.002 *	0.018 *
None (%)	1124 (93)	0.03 (1.31)	0.398 0.391 0.136	883 (82)	0.21 (1.05)	0.621	0.00001 *
Oral regions			0.568			0.004 *	
#14–17 (%)	298 (25)	0.04 (1.21)		302 (28)	0.32 (0.91)		0.00001 *
#24–27 (%)	362 (30)	0.10 (1.50)	0.893	264 (25)	0.07 (1.05)	0.004 *	0.342
#34–37 (%)	274 (23)	−0.03 (1.10)	0.379 0.323	232 (22)	0.18 (0.91)	0.030 0.556	0.024 *
#44–47 (%)	276 (23)	−0.03 (1.29)	0.320 0.265 0.880	273 (25)	0.18 (1.25)	0.001 * 0.768 0.427	0.087
Precise sites			0.735			0.537	
#04 (%)	312 (26)	−0.03 (1.06)		282 (26)	0.11 (0.76)		0.033 *
#05 (%)	312 (26)	−0.04 (0.94)	0.808	282 (26)	0.17 (0.91)	0.180	0.002 *
#06 (%)	306 (25)	0.09 (1.23)	0.291 0.380	261 (24)	0.22 (1.04)	0.198 0.979	0.061
#07 (%)	280 (23)	0.10 (1.84)	0.921 0.961 0.445	246 (23)	0.28 (1.40)	0.394 0.840 0.830	0.039 *

*p* 1: *p* values among subjects in the GTR or Flap groups (*: Statistical significance level was 5%, *p* < 0.05). *p* 2: *p* values between the GTR and Flap groups (*: Statistical significance level was 5%, *p* < 0.05). CBCT interval: Interval between first and second CBCT. #14–17: Right upper premolar and molar region (FDI numbering system). #24–27: Left upper premolar and molar region (FDI numbering system). #34–37: Left lower premolar and molar region (FDI numbering system). #44–47: Right lower premolar and molar region (FDI numbering system). #04: First premolars (FDI numbering system). #05: Second premolars (FDI numbering system). #06: First molars (FDI numbering system). #07: Second molars (FDI numbering system).

**Table 2 medicina-57-00869-t002:** Association of tooth site and bone height between GTR and Flap (mm).

Variable	Surgery				
	GTR		FLAP		*p*
	*N*	Mean (SD)	*N*	Mean (SD)	
Total	1210	0.03 (0.13)	1071	0.19 (0.10)	0.00001 *
Location					
Site #10					
#14M	39	−0.08 (1.38)	38	0.26 (0.67)	0.068
#14D	39	−0.23 (1.21)	38	0.32 (0.73)	0.016 *
#15M	39	0.01 (0.63)	39	0.36 (0.64)	0.036 *
#15D	39	0.36 (1.04)	39	0.30 (0.74)	0.730
#16M	36	0.02 (1.18)	38	0.21 (0.77)	0.584
#16D	36	0.08 (1.39)	38	0.32 (1.17)	0.242
#17M	35	0.04 (0.98)	36	0.40 (1.04)	0.167
#17D	35	0.13 (1.68)	36	0.39 (1.34)	0.060
Site #20					
#24M	48	0.03 (0.98)	36	−0.05 (0.80)	0.978
#24D	48	−0.09 (0.83)	36	−0.13 (0.87)	0.964
#25M	47	−0.22 (1.19)	34	0.14 (0.78)	0.244
#25D	47	−0.29 (1.04)	34	−0.08 (1.33)	0.043 *
#26M	48	0.20 (1.20)	31	0.36 (0.97)	0.778
#26D	48	0.31 (1.62)	31	0.13 (0.88)	0.996
#27M	38	0.92 (2.91)	31	0.05 (1.58)	0.704
#27D	38	0.13 (1.37)	31	0.15 (0.97)	0.587
Site #30					
#34M	33	0.05 (0.98)	31	0.20 (0.64)	0.326
#34D	33	0.04 (0.70)	31	0.14 (0.60)	0.427
#35M	35	−0.22 (0.94)	31	0.17 (0.68)	0.073
#35D	35	0.00 (0.83)	31	0.06 (0.79)	0.867
#36M	35	−0.17 (0.84)	27	−0.01 (0.97)	0.243
#36D	35	0.01 (0.93)	27	0.31 (0.96)	0.203
#37M	34	0.22 (1.41)	27	0.30 (1.31)	0.890
#37D	34	−0.17 (1.81)	27	0.27 (1.23)	0.576
Site #40					
#44M	36	−0.19 (1.01)	36	0.11 (0.98)	0.463
#44D	36	0.24 (1.28)	36	−0.01 (0.60)	0.426
#45M	35	0.16 (0.84)	37	0.08 (0.96)	0.660
#45D	35	−0.02 (0.67)	37	0.25 (1.11)	0.266
#46M	34	0.21 (1.45)	34	0.13 (1.35)	0.628
#46D	34	−0.02 (0.84)	35	0.24 (1.19)	0.601
#47M	33	−0.37 (1.28)	29	0.47 (1.57)	0.023 *
#47D	33	−0.25 (2.29)	29	0.22 (2.08)	0.277

*p*: *p* values among subjects in the GTR or Flap groups (*: Statistical significance level was 5%, *p* < 0.05). GTR: Guided tissue regeneration. #14–17: Right upper premolar and molar region (FDI numbering system). #24–27: Left upper premolar and molar region (FDI numbering system). #34–37: Left lower premolar and molar region (FDI numbering system). #44–47: Right lower premolar and molar region (FDI numbering system).

**Table 3 medicina-57-00869-t003:** Association of sex and bone height between GTR and Flap (mm).

Variable	Sex									
	Male					Female				
	GTR		FLAP		*p*	GTR		FLAP		*p*
	*N*	Mean (SD)	*N*	Mean (SD)		*N*	Mean (SD)	*N*	Mean (SD)	
Total	594	0.00 (0.12)	666	0.18 (0.11)	0.010 *	616	0.05 (0.14)	405	0.20 (0.09)	0.00001 *
Location										
Site #10										
#14M	18	−0.16 (1.86)	20	0.28 (0.80)	0.120	21	−0.01 (0.82)	18	0.25 (0.50)	0.366
#14D	18	−0.08 (0.74)	20	0.25 (0.71)	0.253	21	−0.36 (1.51)	18	0.41 (0.76)	0.028 *
#15M	19	0.02 (0.69)	21	0.19 (0.62)	0.606	20	0.01 (0.58)	18	0.55 (0.62)	0.009 *
#15D	19	0.54 (1.32)	21	0.25 (0.85)	0.635	20	0.20 (0.67)	18	0.36 (0.61)	0.285
#16M	18	0.13 (0.62)	20	0.19 (0.82)	0.769	18	−0.09 (1.56)	18	0.23 (0.73)	0.635
#16D	18	−0.03 (0.89)	20	0.38 (1.25)	0.278	18	0.20 (1.77)	18	0.27 (1.10)	0.569
#17M	17	0.04 (0.69)	18	0.53 (1.23)	0.215	18	0.04 (1.21)	18	0.26 (0.82)	0.410
#17D	17	−0.07 (1.64)	18	0.44 (1.84)	0.203	18	0.31 (1.74)	18	0.34 (0.58)	0.260
Site #20										
#24M	20	0.27 (1.03)	24	−0.14 (0.85)	0.310	28	−0.14 (0.92)	12	0.13 (0.68)	0.214
#24D	20	0.11 (0.77)	24	−0.15 (0.92)	0.315	28	−0.23 (0.85)	12	−0.09 (0.79)	0.594
#25M	20	0.07 (0.77)	23	0.20 (0.83)	0.706	27	−0.43 (1.40)	11	0.03 (0.69)	0.202
#25D	20	−0.16 (0.66)	23	−0.21 (1.55)	0.227	27	−0.40 (1.25)	11	0.20 (0.70)	0.084
#26M	21	0.21 (0.76)	19	0.28 (0.97)	0.734	27	0.19 (1.46)	12	0.48 (1.00)	0.511
#26D	21	0.05 (0.79)	19	0.14 (1.09)	0.776	27	0.50 (2.05)	12	0.12 (0.41)	0.681
#27M	14	0.34 (1.14)	20	0.02 (1.21)	0.612	24	1.26 (3.55)	11	0.12 (2.18)	0.831
#27D	14	−0.11 (1.02)	20	0.19 (1.15)	0.889	24	0.28 (1.54)	11	0.08 (0.56)	0.444
Site #30										
#34M	18	−0.01 (0.93)	22	0.22 (0.70)	0.663	15	0.11 (1.07)	9	0.16 (0.52)	0.323
#34D	18	0.17 (0.76)	22	0.15 (0.56)	0.859	15	−0.11 (0.62)	9	0.12 (0.72)	0.370
#35M	19	−0.18 (1.22)	21	0.06 (0.68)	0.542	16	−0.27 (0.45)	10	0.40 (0.66)	0.012 *
#35D	19	−0.05 (1.03)	21	0.18 (0.93)	0.480	16	0.06 (0.54)	10	−0.19 (0.29)	0.169
#36M	19	−0.18 (1.01)	20	0.08 (0.99)	0.291	16	−0.15 (0.62)	7	−0.27 (0.93)	0.788
#36D	19	−0.02 (0.90)	20	0.46 (1.00)	0.109	16	0.04 (0.98)	7	−0.11 (0.70)	0.867
#37M	17	0.30 (1.51)	19	0.35 (1.42)	0.886	17	0.15 (1.33)	8	0.20 (1.10)	0.815
#37D	17	−0.24 (2.31)	19	0.28 (1.29)	0.775	17	−0.11 (1.20)	8	0.23 (1.14)	0.640
Site #40										
#44M	20	−0.16 (1.15)	24	0.20 (1.12)	0.351	16	−0.23 (0.83)	12	−0.09 (0.62)	0.834
#44D	20	0.26 (1.64)	24	−0.08 (0.64)	0.457	16	0.21 (0.62)	12	0.15 (0.50)	0.889
#45M	19	−0.03 (0.79)	24	−0.03 (1.03)	0.990	16	0.38 (0.88)	13	0.29 (0.79)	0.676
#45D	19	−0.22 (0.66)	24	0.30 (1.33)	0.126	16	0.22 (0.61)	13	0.16 (0.54)	0.947
#46M	19	0.34 (1.82)	23	0.17 (1.58)	0.752	15	0.03 (0.80)	11	0.05 (0.72)	0.658
#46D	19	0.02 (0.69)	23	0.22 (1.41)	0.889	15	−0.06 (1.03)	12	0.28 (0.61)	0.282
#47M	19	−0.77 (1.32)	15	0.39 (1.89)	0.017 *	14	0.19 (1.01)	14	0.56 (1.21)	0.488
#47D	19	−0.33 (2.85)	15	0.60 (2.45)	0.211	14	−0.14 (1.28)	14	−0.19 (1.60)	0.782

*p*: *p* values among subjects in the GTR or Flap groups (*: Statistical significance level was 5%, *p* < 0.05). GTR: Guided tissue regeneration. #14–17: Right upper premolar and molar region (FDI numbering system). #24–27: Left upper premolar and molar region (FDI numbering system). #34–37: Left lower premolar and molar region (FDI numbering system). #44–47: Right lower premolar and molar region (FDI numbering system).

**Table 4 medicina-57-00869-t004:** Association of smoking and bone height between GTR and Flap (mm).

Variable	Smoking														
	Present					Past					Not				
	GTR		Flap		*p*	GTR		Flap		*p*	GTR		Flap		*p*
	*N*	Mean (SD)	*N*	Mean (SD)		*N*	Mean (SD)	*N*	Mean (SD)		*N*	Mean (SD)	*N*	Mean (SD)	
Total	280	0.10 (0.13)	239	0.22 (0.12)	0.120	91	−0.05 (0.11)	82	0.09 (0.11)	0.693	839	0.01 (1.32)	750	0.19 (0.96)	0.009 *
Location															
Site #10															
#14M	8	0.56 (1.87)	4	0.38 (0.34)	0.495	3	−0.13 (0.12)	2	0.15 (0.64)	1.000	28	−0.25 (1.27)	32	0.26 (0.71)	0.096
#14D	10	−0.03 (0.63)	9	0.33 (0.41)	0.205	5	0.16 (0.45)	3	0.63 (0.67)	0.131	24	−0.39 (1.46)	26	0.28 (0.82)	0.080
#15M	9	−0.11 (0.52)	6	0.13 (0.29)	0.473	4	−0.58 (0.74)	1	N/A	0.277	26	0.15 (0.60)	32	0.40 (0.69)	0.194
#15D	10	0.15 (0.95)	11	0.47 (1.08)	0.501	5	1.46 (2.07)	1	N/A	0.546	24	0.23 (0.62)	27	0.22 (0.57)	0.769
#16M	5	−0.66 (1.30)	5	0.02 (1.21)	0.465	3	0.63 (0.50)	1	N/A	0.180	28	0.08 (1.18)	32	0.25 (0.71)	0.630
#16D	8	0.29 (0.38)	11	0.17 (0.78)	0.589	6	−0.07 (0.58)	2	0.40 (0.14)	0.317	22	0.05 (1.75)	25	0.38 (1.35)	0.276
#17M	3	−0.13 (0.95)	6	0.48 (0.59)	0.362	1	N/A	1	N/A	0.317	31	0.10 (0.98)	29	0.38 (1.13)	0.366
#17D	9	0.82 (1.92)	10	0.88 (2.07)	0.838	3	−0.50 (0.52)	4	−0.23 (0.30)	0.368	23	−0.06 (1.64)	22	0.28 (0.97)	0.044
Site #20															
#24M	9	0.69 (1.27)	6	−0.43 (0.60)	0.037*	1	N/A	0	N/A	N/A					
#24D	12	0.36 (0.45)	14	−0.34 (0.88)	0.074	5	−0.32 (0.99)	3	−1.03 (1.30)	0.294	38	−0.13 (0.86)	30	0.02 (0.82)	0.332
#25M	8	0.10 (0.62)	7	−0.11 (0.89)	0.602	1	N/A (N/A)	1	N/A	0.317	31	−0.23 (0.87)	19	0.17 (0.66)	0.126
#25D	14	−0.09 (0.74)	9	0.12 (0.69)	0.229	4	−0.83 (0.76)	5	0.48 (0.53)	0.027*	38	−0.26 (1.28)	26	0.25 (0.74)	0.089
#26M	10	−0.26 (0.80)	6	−0.07 (0.70)	0.785	2	0.15 (0.21)	4	0.53 (1.36)	1.000	29	−0.32 (1.18)	20	−0.31 (1.64)	0.380
#26D	10	−0.08 (0.80)	9	−0.17 (1.17)	0.870	5	0.02 (0.18)	3	0.80 (0.96)	0.453	36	0.33 (1.29)	21	0.44 (0.97)	0.993
#27M	9	1.24 (1.49)	5	−1.06 (1.31)	0.019 *	1	N/A	3	0.67 (0.65)	0.180	33	0.47 (1.90)	19	0.17 (0.67)	0.754
#27D	14	0.46 (1.48)	13	0.34 (1.30)	0.697	1	N/A	4	−0.03 (0.41)	0.717	28	0.94 (3.24)	23	0.21 (1.65)	0.622
Site #30															
#34M	6	0.10 (0.73)	3	−0.37 (0.67)	0.362	2	−0.20 (2.12)	3	−0.07 (0.55)	1.000	25	0.06 (0.99)	25	0.3 (0.63)	0.161
#34D	11	0.04 (0.65)	8	0.23 (0.52)	0.617	2	−0.40 (0.28)	2	−0.35 (0.64)	1.000	20	0.09 (0.76)	21	0.15 (0.63)	0.522
#35M	6	−1.05 (1.23)	2	−0.10 (0.14)	0.129	1	N/A	1	N/A	0.317	28	−0.07 (0.80)	28	0.16 (0.69)	0.221
#35D	12	−0.18 (1.16)	10	0.36 (0.55)	0.289	2	0.75 (0.07)	2	0.55 (1.20)	1.000	21	0.03 (0.60)	19	−0.15 (0.83)	0.635
#36M	4	−0.10 (0.22)	4	−0.18 (1.98)	0.468	2	−0.50 (0.14)	1	N/A	0.480	29	−0.16 (0.92)	22	0.05 (0.76)	0.336
#36D	10	−0.05 (0.75)	9	0.37 (0.97)	0.487	3	0.57 (0.90)	3	0.20 (1.13)	0.827	22	−0.04 (1.01)	15	0.29 (0.99)	0.233
#37M	7	0.59 (1.51)	4	1.48 (2.13)	0.448	3	0.50 (0.50)	4	0.33 (1.30)	0.858	24	0.08 (1.46)	19	0.05 (1.04)	0.961
#37D	9	−0.51 (2.84)	6	0.22 (0.93)	0.637	2	0.70 (0.99)	4	−1.20 (0.37)	0.064	23	−0.12 (1.35)	17	0.63 (1.21)	0.118
Site #40															
#44M	4	0.00 (0.29)	8	0.59 (1.81)	0.495	3	−0.40 (0.80)	2	0.10 (0.28)	0.564	29	−0.20 (1.10)	26	−0.04 (0.57)	0.800
#44D	10	1.12 (1.93)	10	−0.01 (0.55)	0.041 *	3	−1.00 (1.28)	4	−0.60 (1.16)	0.724	23	0.01 (0.55)	22	0.10 (0.45)	0.715
#45M	5	−0.50 (0.57)	5	0.84 (1.06)	0.028 *	2	0.75 (1.06)	3	−0.03 (1.50)	0.564	28	0.23 (0.83)	29	−0.04 (0.86)	0.143
#45D	9	−0.03 (0.64)	10	0.02 (0.77)	0.652	3	−0.17 (1.11)	6	0.88 (2.27)	0.362	23	0.00 (0.65)	21	0.18 (0.71)	0.409
#46M	7	−0.03 (0.79)	5	−0.24 (2.10)	0.569	1	N/A	2	0.85 (3.89)	1.000	26	0.24 (1.60)	27	0.14 (1.00)	0.624
#46D	15	−0.37 (0.93)	12	0.63 (1.72)	0.117	3	0.20 (0.36)	3	−0.37 (0.49)	0.127	16	0.27 (0.73)	20	0.10 (0.78)	0.678
#47M	6	−0.43 (1.01)	6	0.18 (1.08)	0.226	4	−0.15 (0.34)	1	N/A	0.480	23	−0.39 (1.45)	22	0.59 (1.72)	0.058
#47D	11	0.25 (2.75)	6	1.40 (3.26)	0.392	5	−0.70 (1.98)	3	−0.63 (0.60)	1.000	17	−0.44 (2.13)	20	−0.01 (1.72)	0.234

*p*: *p* values among subjects in the GTR or Flap groups (*: Statistical significance level was 5%, *p* < 0.05). GTR: Guided tissue regeneration. #14–17: Right upper premolar and molar region (FDI numbering system). #24–27: Left upper premolar and molar region (FDI numbering system). #34–37: Left lower premolar and molar region (FDI numbering system). #44–47: Right lower premolar and molar region (FDI numbering system).

**Table 5 medicina-57-00869-t005:** Association of age and bone height between GTR and Flap (mm).

Variable	Age														
	29–45 Years Old					46–53 Years Old					54–76 Years Old				
	GTR		FLAP		*p*	GTR		FLAP		*p*	GTR		FLAP		*p*
	*N*	Mean (SD)	*N*	Mean (SD)		*N*	Mean (SD)	*N*	Mean (SD)		*N*	Mean (SD)	*N*	Mean (SD)	
Total	416	−0.17 (0.12)	376	0.17 (0.10)	0.00001 *	388	0.19 (0.13)	312	0.24 (0.11)	0.040 *	406	0.08 (0.14)	383	0.16 (0.10)	0.125
Location															
Site #10															
#14M	10	−0.23 (0.55)	13	0.19 (0.39)	0.087	13	−0.02 (0.46)	10	0.12 (0.78)	0.618	16	−0.03 (2.11)	15	0.43 (0.78)	0.352
#14D	10	−0.32 (0.66)	13	0.26 (0.41)	0.016 *	13	0.03 (0.55)	10	0.43 (1.06)	0.223	16	−0.38 (1.76)	15	0.31 (0.72)	0.332
#15M	11	−0.03 (0.56)	13	0.22 (0.51)	0.337	12	−0.01 (0.72)	10	0.34 (0.52)	0.186	16	0.06 (0.63)	16	0.48 (0.80)	0.162
#15D	11	0.66 (1.71)	13	0.19 (0.67)	0.642	12	0.18 (0.56)	10	0.34 (0.62)	0.427	16	0.30 (0.67)	16	0.36 (0.88)	0.777
#16M	9	−0.71 (1.10)	13	0.14 (0.67)	0.076	12	0.48 (0.77)	9	0.30 (1.03)	0.943	15	0.09 (1.33)	16	0.21 (0.73)	0.905
#16D	9	−0.32 (1.24)	13	0.20 (0.92)	0.192	12	0.06 (0.76)	9	0.78 (1.53)	0.251	15	0.35 (1.82)	16	0.17 (1.13)	0.782
#17M	8	−0.58 (1.30)	13	0.39 (1.37)	0.180	12	0.1 (0.63)	7	0.63 (0.51)	0.074	15	0.32 (0.92)	16	0.30 (0.94)	0.766
#17D	8	−0.43 (1.02)	13	0.35 (1.83)	0.514	12	0.67 (2.44)	7	0.06 (0.57)	1.000	15	−0.01 (1.10)	16	0.56 (1.14)	0.038
Site #20															
#24M	16	0.03 (1.04)	14	−0.05 (0.76)	0.532	17	0.03 (0.76)	11	0.02 (1.08)	0.981	15	0.03 (1.19)	11	−0.13 (0.57)	0.549
#24D	16	0.13 (0.53)	14	−0.11 (0.67)	0.631	17	0.08 (0.73)	11	0.36 (0.87)	0.239	15	−0.52 (1.05)	11	−0.64 (0.87)	0.585
#25M	15	−0.29 (1.76)	14	0.07 (0.6)	0.930	17	−0.08 (0.55)	10	0.40 (0.98)	0.174	15	−0.3 (1.08)	10	−0.01 (0.81)	0.469
#25D	15	−0.39 (1.47)	14	−0.08 (0.93)	0.457	17	−0.13 (0.73)	10	−0.11 (2.23)	0.050 *	15	−0.39 (0.85)	10	−0.04 (0.56)	0.502
#26M	16	−0.21 (0.94)	10	0.08 (0.58)	0.544	17	0.17 (0.74)	9	0.39 (1.02)	0.808	15	0.67 (1.67)	12	0.56 (1.19)	0.825
#26D	16	−0.35 (1.14)	10	0.11 (0.81)	0.398	17	0.29 (0.79)	9	0.34 (0.83)	0.646	15	1.02 (2.39)	12	−0.01 (1.01)	0.170
#27M	11	−0.57 (1.39)	12	0.03 (1.08)	0.281	14	1.84 (3.48)	9	0.61 (1.34)	0.614	13	1.20 (2.89)	10	−0.43 (2.18)	0.238
#27D	11	−0.32 (1.05)	12	−0.11 (1.39)	0.711	14	0.06 (1.26)	9	0.60 (0.50)	0.072	13	0.60 (1.64)	10	0.05 (0.50)	0.071
Site #30															
#34M	16	−0.07 (0.67)	11	0.16 (0.49)	0.308	10	0.49 (1.17)	9	−0.03 (0.77)	0.436	7	−0.31 (1.19)	11	0.43 (0.65)	0.220
#34D	16	0.05 (0.66)	11	0.06 (0.52)	0.921	10	0.09 (0.53)	9	0.43 (0.55)	0.175	7	−0.04 (1.05)	11	−0.03 (0.67)	0.555
#35M	17	−0.29 (0.68)	11	0.36 (0.78)	0.036 *	11	−0.20 (0.53)	9	−0.04 (0.65)	0.760	7	−0.09 (1.80)	11	0.16 (0.6)	0.496
#35D	17	−0.03 (0.71)	11	0.36 (0.58)	0.229	11	0.15 (0.6)	9	−0.27 (0.82)	0.237	7	−0.16 (1.38)	11	0.04 (0.91)	0.964
#36M	17	−0.26 (0.62)	9	−0.4 (1.36)	0.552	11	0.00 (0.67)	8	0.19 (0.68)	0.455	7	−0.21 (1.46)	10	0.18 (0.70)	0.961
#36D	17	−0.08 (0.83)	9	0.36 (0.58)	0.124	11	−0.04 (1.02)	8	0.29 (1.11)	0.508	7	0.30 (1.08)	10	0.28 (1.17)	0.660
#37M	18	0.02 (1.58)	9	−0.07 (1.25)	0.837	9	0.50 (1.14)	7	−0.37 (0.83)	0.090	7	0.40 (1.32)	11	1.04 (1.34)	0.585
#37D	18	−0.24 (2.14)	9	0.36 (1.21)	0.381	9	0.2 (1.55)	7	−0.63 (0.57)	0.080	7	−0.49 (1.22)	11	0.76 (1.31)	0.146
Site #40															
#44M	12	−0.14 (1.19)	12	0.25 (0.55)	0.469	10	0.01 (1.03)	13	0.32 (1.47)	0.755	14	−0.38 (0.85)	11	−0.31 (0.39)	0.600
#44D	12	0.37 (2.06)	12	0.23 (0.47)	0.469	10	0.10 (0.35)	13	0.06 (0.72)	0.619	14	0.22 (0.83)	11	−0.35 (0.45)	0.061
#45M	11	0.33 (0.75)	12	0.24 (0.83)	0.643	10	0.25 (0.74)	13	0.22 (0.96)	0.828	14	−0.04 (0.99)	12	−0.23 (1.07)	0.680
#45D	11	0.12 (0.57)	12	0.38 (0.36)	0.153	10	−0.18 (0.74)	13	0.58 (1.71)	0.144	14	−0.01 (0.70)	12	−0.23 (0.56)	0.353
#46M	11	0.24 (0.72)	12	−0.02 (1.13)	0.478	10	0.36 (2.17)	13	0.37 (1.75)	0.153	13	0.06 (1.33)	9	−0.03 (1.03)	0.947
#46D	11	0.19 (0.96)	12	0.13 (1.39)	0.579	10	0.15 (0.60)	13	0.39 (1.34)	0.733	13	−0.32 (0.87)	10	0.19 (0.71)	0.250
#47M	10	−1.05 (1.64)	10	0.69 (2.12)	0.010 *	9	−0.10 (1.15)	9	−0.09 (1.16)	0.965	14	−0.05 (0.89)	10	0.76 (1.25)	0.150
#47D	10	−1.64 (2.05)	10	0.77 (2.64)	0.011 *	9	0.88 (3.29)	9	0.28 (1.61)	0.626	14	0.02 (0.95)	10	−0.39 (1.86)	0.953

*p*: *p* values among subjects in the GTR or Flap groups (*: Statistical significance level was 5%, *p* < 0.05). GTR: Guided tissue regeneration. #14–17: Right upper premolar and molar region (FDI numbering system). #24–27: Left upper premolar and molar region (FDI numbering system). #34–37: Left lower premolar and molar region (FDI numbering system). #44–47: Right lower premolar and molar region (FDI numbering system).

**Table 6 medicina-57-00869-t006:** Association of CBCT interval and bone height between GTR and Flap (mm).

Variable	CBCT Interval														
	87–490 Days					491–859 Days					860–1543 Days				
	GTR		Flap		*p*	GTR		Flap		*p*	GTR		Flap		*p*
	*N*	Mean (SD)	*N*	Mean (SD)		*N*	Mean (SD)	*N*	Mean (SD)		*N*	Mean (SD)	*N*	Mean (SD)	
Total	404	−0.06 (0.12)	332	0.17 (0.08)	0.00001 *	362	0.02 (0.12)	419	0.20 (0.12)	0.018 *	444	0.11 (0.14)	320	0.20 (0.10)	0.092
Location															
Site #10															
#14M	11	−0.30 (0.91)	14	0.20 (0.51)	0.084	12	0.10 (0.56)	18	0.33 (0.85)	0.308	16	−0.06 (2.00)	6	0.20 (0.34)	0.529
#14D	11	−0.06 (0.70)	14	0.54 (0.85)	0.094	12	−0.73 (1.94)	18	0.26 (0.67)	0.048 *	16	0.04 (0.56)	6	0.02 (0.52)	0.912
#15M	11	−0.14 (0.62)	14	0.26 (0.63)	0.146	12	−0.05 (0.71)	19	0.46 (0.71)	0.084	16	0.16 (0.58)	6	0.23 (0.45)	0.767
#15D	11	−0.06 (0.63)	14	0.33 (0.76)	0.138	12	0.54 (0.78)	19	0.31 (0.82)	0.502	16	0.53 (1.35)	6	0.20 (0.46)	0.795
#16M	11	−0.11 (1.29)	14	0.38 (0.74)	0.285	11	0.23 (1.32)	17	0.23 (0.77)	0.869	14	−0.04 (1.02)	7	−0.19 (0.79)	0.525
#16D	11	−0.44 (1.23)	14	0.14 (0.74)	0.188	11	0.50 (2.05)	17	0.64 (1.54)	0.247	14	0.16 (0.65)	7	−0.07 (0.63)	0.501
#17M	9	−0.26 (1.12)	12	0.57 (0.71)	0.039 *	11	0.23 (0.81)	18	0.42 (1.17)	0.770	15	0.08 (1.02)	6	0.00 (1.24)	0.969
#17D	9	−0.01 (0.98)	12	0.38 (1.02)	0.545	11	0.50 (2.18)	18	0.32 (1.73)	0.787	15	−0.06 (1.65)	6	0.60 (0.36)	0.017 *
Site #20															
#24M	18	−0.19 (0.50)	13	0.10 (0.69)	0.074	12	0.13 (0.93)	11	0.14 (0.95)	0.734	18	0.18 (1.33)	12	−0.39 (0.71)	0.122
#24D	18	−0.04 (0.66)	13	−0.36 (0.92)	0.470	12	0.19 (0.61)	11	−0.03 (1.15)	0.926	18	−0.33 (1.04)	12	0.03 (0.39)	0.339
#25M	17	−0.42 (1.55)	13	0.08 (0.86)	0.502	11	−0.04 (0.71)	10	0.20 (0.96)	0.502	19	−0.14 (1.07)	11	0.17 (0.54)	0.289
#25D	17	−0.40 (1.33)	13	0.07 (0.71)	0.130	11	−0.28 (0.68)	10	−0.47 (2.24)	0.306	19	−0.21 (0.95)	11	0.11 (0.73)	0.342
#26M	18	0.39 (1.69)	10	0.10 (0.65)	0.386	12	−0.21 (0.74)	10	0.72 (0.96)	0.032 *	18	0.27 (0.75)	11	0.26 (1.18)	0.557
#26D	18	0.53 (1.86)	10	0.25 (0.56)	0.665	12	0.26 (1.94)	10	0.11 (1.33)	0.575	18	0.11 (1.14)	11	0.05 (0.65)	0.840
#27M	12	0.93 (1.85)	10	−0.18 (0.99)	0.261	10	1.14 (3.03)	10	−0.27 (2.42)	0.384	16	0.79 (3.59)	11	0.56 (0.98)	0.236
#27D	12	0.09 (1.73)	10	−0.11 (0.80)	0.766	10	−0.04 (1.00)	10	0.12 (0.76)	0.705	16	0.28 (1.33)	11	0.41 (1.26)	0.225
Site #30															
#34M	10	−0.43 (0.72)	7	0.16 (0.49)	0.095	13	0.16 (0.50)	12	0.13 (0.79)	0.445	10	0.38 (1.47)	12	0.30 (0.59)	0.947
#34D	10	−0.11 (0.40)	7	0.26 (0.32)	0.077	13	0.15 (0.64)	12	0.08 (0.50)	0.784	10	0.06 (1.00)	12	0.13 (0.81)	0.869
#35M	10	0.05 (0.22)	8	−0.05 (0.78)	0.591	15	−0.28 (0.71)	12	0.33 (0.65)	0.042 *	10	−0.41 (1.53)	11	0.16 (0.66)	0.191
#35D	10	−0.13 (0.63)	8	0.29 (0.49)	0.180	15	0.31 (0.56)	12	−0.01 (0.91)	0.261	10	−0.33 (1.20)	11	−0.03 (0.86)	0.572
#36M	11	−0.18 (0.81)	6	−0.10 (1.11)	0.480	14	−0.10 (0.49)	11	−0.47 (0.92)	0.364	10	−0.25 (1.25)	10	0.55 (0.70)	0.049 *
#36D	11	0.13 (0.78)	6	0.22 (0.59)	0.920	14	−0.21 (0.82)	11	0.42 (1.23)	0.198	10	0.20 (1.20)	10	0.24 (0.85)	0.677
#37M	9	0.63 (1.62)	5	−0.30 (0.96)	0.255	15	0.01 (1.35)	11	−0.02 (0.82)	0.856	10	0.18 (1.36)	11	0.90 (1.66)	0.307
#37D	9	−0.74 (2.63)	5	0.14 (0.77)	0.503	15	0.41 (1.11)	11	0.38 (1.54)	0.795	10	−0.53 (1.71)	11	0.21 (1.14)	0.230
Site #40															
#44M	14	0.01 (0.55)	11	−0.04 (0.30)	0.659	9	−0.64 (1.41)	13	0.58 (1.42)	0.116	13	−0.10 (1.05)	12	−0.28 (0.58)	0.785
#44D	14	0.06 (0.53)	11	−0.02 (0.53)	0.783	9	−0.44 (0.92)	13	−0.06 (0.76)	0.547	13	0.90 (1.74)	12	0.07 (0.51)	0.091
#45M	14	0.14 (0.95)	10	0.03 (0.70)	0.837	9	−0.24 (0.81)	14	−0.06 (0.84)	0.924	12	0.48 (0.63)	13	0.28 (1.25)	0.978
#45D	14	−0.11 (0.86)	10	0.61 (1.74)	0.394	9	−0.08 (0.48)	14	0.03 (0.61)	0.567	12	0.13 (0.53)	13	0.21 (0.93)	0.585
#46M	14	−0.34 (1.01)	11	0.42 (1.39)	0.227	8	0.33 (1.29)	11	0.15 (1.07)	1.000	12	0.77 (1.81)	12	−0.16 (1.59)	0.339
#46D	14	−0.06 (1.00)	11	0.27 (1.50)	0.913	8	−0.01 (0.92)	12	0.17 (1.32)	0.817	12	0.03 (0.65)	12	0.29 (0.76)	0.418
#47M	13	−0.25 (0.95)	8	0.21 (0.49)	0.076	7	−1.14 (2.01)	12	0.36 (2.11)	0.117	13	−0.07 (0.97)	9	0.86 (1.44)	0.150
#47D	13	−0.34 (2.48)	8	0.06 (0.61)	0.514	7	−0.87 (1.42)	12	−0.01 (2.93)	0.525	13	0.18 (2.53)	9	0.66 (1.66)	0.332

*p*: *p* values among subjects in the GTR or Flap groups (*: Statistical significance level was 5%, *p* < 0.05). GTR: Guided tissue regeneration. #14–17: Right upper premolar and molar region (FDI numbering system). #24–27: Left upper premolar and molar region (FDI numbering system). #34–37: Left lower premolar and molar region (FDI numbering system). #44–47: Right lower premolar and molar region (FDI numbering system).

**Table 7 medicina-57-00869-t007:** Multivariable analysis association between alveolar bone loss and other variables after treatment.

	Alveolar Bone Loss		
	*OR*	95% CI	*p*
Type of surgery (GTR/Flap)	0.731	0.619–0.862	0.001 *
Sex	1.025	0.849–1.237	0.798
Age	0.998	0.988–1.009	0.753
Smoking	0.979	0.790–1.214	0.847
Systemic diseases	1.275	0.794–2.049	0.354
Medicine	1.295	0.804–2.085	0.514
Upper/Lower jaw	1.255	1.063–1.481	0.007 *
Sites	1.055	0.832–1.336	0.528

Multivariable analysis using logistic regression model: type of surgery, sex, age, smoking, systemic disease, medicine, upper/lower jaw, and precise sites (*: Statistical significance level was 5%, *p* < 0.05).

## Data Availability

All available data are presented within the article or are available on request from the corresponding author.

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
