# Peer review of "Evaluating Change of Marginal Bone Height with Cone-Beam Computed Tomography Following Surgical Treatment with Guided Tissue Regeneration (Bone Grafting) or Access Flap Alone: A Retrospective Study"

_medicina, 2021, doi:10.3390/medicina57090869_

Round 1

Reviewer 1 Report

I believe this study presents a great insight about the efficacy of guided tissue regeneration x access flap alone. However, the methodology still needs to be clarified. Please see my questions and suggestions below.

1.How the sample size was defined?

2.Which parameters were used to define chronic periodontitis?

3.Who and how many persons measured the CBCTs? Is there was any calibration to attest the consistency of the measurements?

4.It would be great to have a summary about the association of smoking and other variables to the treatment option. The authors could use the median/average of measurements to perform a regression analysis or an ANOVA/Kruskal considering the co-variants.

5.Limitations of this study and clinical significance shroud be further explored in the Discussion section.

Author Response

Thanks for your comment. We did our best to satisfy your comments. I attach a revision note and revised manuscript including figures and table for you. Thanks again for your kindness.

Sincerely, 

Jung-Tae Lee

Reviewer 2 Report

Dear authors, I recommend your article to be published after following minor revisions:

row 19: Change "flap" to "Flap".

row 23: P=0.000 looks strange, change it to real number, i.e. 0.00006 or 5*10-6.

row 51: Change "procedures" to "procedure".

row 79: Change "assessment., CBCT" to "assessment. CBCT".

Figure 1: Change three times "FLAP" to "Flap" and "CBCT images without before and after" to "without CBCT images before and after".

row 164: Change "a and d" to "A and D".

row 170: I do not understand what "ae" means. Explain it and do not use unexplained abbreviation.

row 177: P=0.000 looks strange, change it to real number, i.e. 0.00006 or 5*10-6.

row 178-9: Your numbers and percentages are not correct here. According to Table 1, it should be corrected as follows: Change "60.0%" to "49.1%", change "40.0%" to "50.9%", change "566" to "666", change "55.0%" to "62.2%", change "616" to "405" and change "45.0%" to "37.8%".

row 180: P=0.000 looks strange, change it to real number, i.e. 0.00006 or 5*10-6.

rows 181, 187: For better reading of the text, round the values to one decimal place.

rows 182, 184, 191, 197, 201, 204: P=0.000 looks strange, change it to real number, i.e. 0.00006 or 5*10-6.

rows 185-6, 189-90, 194, 196-7, 199, 201, 203-4, 205-6: For better reading of the text, round the values to two decimal places.

row 195: Specify "other diseases".

row 201: Delete "results the".

Table 1:

Change "FLAP" to "Flap".

Your percentages in males and females are not correct. Change "60.0%" to "49.1%", change "40.0%" to "50.9%", change "55.0%" to "62.2%" and change "45.0%" to "37.8%".

In part Smoking, GTR group, if I count together Present (280)+Past (91)+Not (398), it is 769, but total number is 1210. What about the difference? It should be also included in these 3 categories (Present, past and not). Am I right? Please, edit this category-Smoking-GTR- so that it is correct. The same for Flap group, 239+82+334=655, but total number is 1071. Please, edit this category-Smoking-Flap- so that it is correct.

rows 239-40: Delete "between".

rows 252-5, 257-9: For better reading of the text, round the values to two decimal places.

row 261: Delete " The GTR and Flap groups by smoking status.".

row 274: Add "of" between "Association" and "CBCT".

Table 2:

Change title "Association bone......" to "Association of tooth site and bone height between GTR and Flap (mm)".

Change "FLAP" to "Flap".

For better reading and clarity, round all values (mean (SD)) to two decimal places.

In first row (Total), P=0.000 looks strange, change it to real number, i.e. 0.00006 or 5*10-6.

Table 3:

Change title "Association bone......" to "Association of sex and bone height between GTR and Flap (mm)".

Change "FLAP" to "Flap".

For better reading and clarity, round all values (mean (SD)) to two decimal places.

In first row (Total), P=0.000 looks strange, change it to real number, i.e. 0.00006 or 5*10-6.

Table 4:

Change title "Association bone......" to "Association of smoking and bone height between GTR and Flap (mm)".

Similarly as my comment to Table 1, in GTR, if I count together Present (280)+Past (91)+Not (398), it is 769, but total number should be 1210. What about the difference? It should be also included in these 3 categories (Present, past and not). Am I right? Please, edit this category-Smoking-GTR- so that it is correct. The same for Flap group, 239+82+334=655, but total number should be 1071. Please, edit this category-Smoking-Flap- so that it is correct.

Change "FLAP" to "Flap".

For better reading and clarity, round all values (mean (SD)) to two decimal places.

Table 5:

Change title "Association bone......" to "Association of age and bone height between GTR and Flap (mm)".

Change "FLAP" to "Flap".

For better reading and clarity, round all values (mean (SD)) to two decimal places.

In first row (Total), P=0.000 looks strange, change it to real number, i.e. 0.00006 or 5*10-6.

Table 6:

Change title "Association bone......" to "Association of CBCT interval and bone height between GTR and Flap (mm)".

Change "FLAP" to "Flap".

For better reading and clarity, round all values (mean (SD)) to two decimal places.

In first row (Total), P=0.000 looks strange, change it to real number, i.e. 0.00006 or 5*10-6.

Discussion-row 8: Delete "3".

Discussion-row 60: Add "situation" between "compared" and "before".

Discussion-rows 77-9: Delete this sentence as it is the same as in Conclusions.

Author Response

(The authors gave the same response as above.)

Round 2

Reviewer 1 Report

Thank you for the answers. I do not have any other comments at this time.

Author Response

I'm very pleased for your respond. thanks again